# Colitis and Colorectal Carcinogenesis: The Focus on Isolated Lymphoid Follicles

**DOI:** 10.3390/biomedicines10020226

**Published:** 2022-01-21

**Authors:** Györgyi Műzes, Bettina Bohusné Barta, Ferenc Sipos

**Affiliations:** Department of Internal Medicine and Hematology, Semmelweis University, 1088 Budapest, Hungary; muzes.gyorgyi@med.semmelweis-univ.hu (G.M.); bettina.barta@gmail.com (B.B.B.)

**Keywords:** subepithelial compartment, isolated lymphoid follicles, tertiary lymphoid organs, colitis, colorectal cancer

## Abstract

Gut-associated lymphoid tissue is one of the most diverse and complex immune compartments in the human body. The subepithelial compartment of the gut consists of immune cells of innate and adaptive immunity, non-hematopoietic mesenchymal cells, and stem cells of different origins, and is organized into secondary (and even tertiary) lymphoid organs, such as Peyer’s patches, cryptopatches, and isolated lymphoid follicles. The function of isolated lymphoid follicles is multifaceted; they play a role in the development and regeneration of the large intestine and the maintenance of (immune) homeostasis. Isolated lymphoid follicles are also extensively associated with the epithelium and its conventional and non-conventional immune cells; hence, they can also function as a starting point or maintainer of pathological processes such as inflammatory bowel diseases or colorectal carcinogenesis. These relationships can significantly affect both physiological and pathological processes of the intestines. We aim to provide an overview of the latest knowledge of isolated lymphoid follicles in colonic inflammation and colorectal carcinogenesis. Further studies of these lymphoid organs will likely lead to an extended understanding of how immune responses are initiated and controlled within the large intestine, along with the possibility of creating novel mucosal vaccinations and ways to treat inflammatory bowel disease or colorectal cancer.

## 1. Introduction

The gastrointestinal system has a well-organized, advanced immune network, because it is frequently assaulted by foreign antigens. In the colon, inflammatory ulceration or carcinoma formation can occur if the balance between the division and breakdown of the epithelial cell layer is disrupted. The integrity of the colonic mucosa is preserved by the complex interplay of immunological activities affecting both the epithelial layer and the subepithelial compartment.

In the past decade, we have learned not only about the anatomical structure and cell composition of the subepithelial compartment of the gut-associated lymphoid tissue (GALT), but thanks to rapidly evolving technical possibilities, we have also accumulated much new knowledge about the immunological functions of GALT, its interaction with the intestinal microbiome, and the maintenance of immune homeostasis. Several new details have also become known about the role of isolated lymphoid follicles (ILFs) in the development of pathological conditions, such as colonic inflammation, or sporadic or colitis-associated colorectal carcinogenesis.

The purpose of this review is to summarize this new knowledge in certain respects and outline future opportunities.

## 2. Organization and Commensal Bacteria of the Gut-Associated Lymphoid Tissue

MALT represents the largest lymphoid compartment within the human body [1]. The MALT’s innate and adaptive components distinguish pathogens from commensal bacteria. Based on anatomic location, MALT is classified into several subgroups, such as bronchus-associated, conjunctiva-associated, gut-associated, larynx-associated, and nose-associated lymphoid tissues [2]. Skin-associated lymphoid tissue [3,4] and vascular-associated lymphoid tissue [5,6] are other distinct types of lymphoid tissue (Figure 1).

GALT is made up of multifollicular isolated and aggregated lymphoid formations that are distributed in the small and large intestines’ subepithelial compartments [7,8,9]. The lymphoid follicles—which include Peyer’s patches (PP), the vermiform appendix, ILFs, cryptopatches (CPs), and lymphoid aggregates (LAs)—consist of a unique follicle-associated epithelium (FAE) that covers the subepithelial dome, where a variety of immune cells (e.g., dendritic cells, macrophages, T and B lymphocytes) are located [7,9,10].

M (microfold) cells—the special antigen-presenting cells of the FAE—transport samples from luminal substances directly to intraepithelial and subepithelial lymphoid cells via transepithelial vesicular transfer, allowing immune responses to occur against intestinal antigens/pathogens [7,11,12]. M cells are likely to be the first to process most intestinal antigens [12,13].

Human PPs are located on the anti-mesenteric side of the small intestine, and have 10–100 distinct follicles [9]. In the human intestine, ILFs have an average diameter of 0.1–1.3 mm and a population of 30,000 [14,15,16]. ILFs have recently been classified into two types, i.e., mucosal ILF (M-ILF) and submucosal ILF (SM-ILF) [9]. M-ILFs are found only in the mucosal lamina propria (LP), whereas SM-ILFs protrude into the LP and present in the submucosa. M-ILFs are mostly located in the ileum and the distal colon, whereas SM-ILFs are found mainly in the colon, and only infrequently in the ileum [9].

As a mutualistic ecosystem, the gastrointestinal commensal microbiota plays multiple roles in preserving host health and triggering host disease. Dysbiosis of the gut microbes can result in inflammation, barrier breakdown, mucosal damage, and an altered microenvironment conducive to the development of colon cancer [17]. In healthy mammals, the underlying region of the intestinal epithelium was formerly assumed to be sterile. Despite the fact that pathogenic bacteria can penetrate the inner mucus and translocate across the intestinal epithelium, lamina propria macrophages and other lymphoid cells in the GALT can promptly kill them. Recent research suggests, however, that different types of commensal bacteria (i.e., *Lactobacillus* spp., and segmented filamentous bacteria) populate the surface of secondary lymphoid organs, and other types of commensal bacteria (i.e., *Alcaligenes* spp. and *Ochrobactrum* spp.) can colonize and proliferate in the GALT of healthy mammals (including humans) by consuming resources from lymphoid tissue [17]. Furthermore, lymphoid-tissue-resident commensal bacteria (i.e., *Achromobacter* spp., *Bordetella* spp., *Ochrobactrum* spp., *Serratia* spp., *Alcaligenes* spp., and *Pseudomonas* spp.) are distinct from lumen-resident bacteria (i.e., *Bacteroides*, *Prevotella*, *Mucispirillum*, *Lactobacillus*, *Ruminococcus*, *Oscillospira*, *Sutterella*, *Desulfovibrio*, and *Fusobacterium*) and epithelium-associated bacteria (i.e., adherent-invasive *Escherichia coli*, segmented filamentous bacteria, *Enterococcus faecalis*, *Bacteroides fragilis*, and *Clostridium* spp.) [17].

## 3. Innate Lymphoid Cells of the Subepithelial Compartment

Innate lymphoid cells (ILCs) are essential for the formation and development of the subepithelial lymphoid compartment. ILCs are evolutionarily ancient cells that provide the primitive immune system with the ability to respond properly to pathogens [18]. These cells are part of the innate immune response, but they share a common lymphoid progenitor with lymphocytes. As a result, while ILCs are the innate counterparts of T cells, they lack T-cell receptors produced by somatic recombination of antigen-specific receptors. ILCs respond quickly to signals or cytokines produced by other cells, allowing them to act early in the immune response [19]. Other distinguishing features of ILCs from lymphocytes include the fact that, unlike intestinal B, T, and NK cells, their intestinal population is not constantly replaced from the circulation, and the ILCs constantly produce their representative cytokines at a steady rate, in contrast to T cells’ on-demand production [19,20].

Human ILCs are divided into cytolytic (e.g., conventional NK cells) or non-cytolytic cells [21,22,23,24,25]. Non-cytolytic ILCs are further classified into ILC1, ILC2, and ILC3 T-helper subsets based on cytokine and transcription factor expression [25,26,27,28,29]. ILC1 cells represent a phenotypically diverse group of tissue-resident cells in the intestine that generate type 1 cytokines, such as interferon (IFN)-γ, and require T-box expressed in T cells (T-BET) expression [29,30]. ILC2 cells can be found both in lymphoid and non-lymphoid tissues, including the intestines. ILC2s synthesize type 2 cytokines (e.g., IL-5, IL-13) and express GATA-3 (transcription factors binding to the DNA sequence "GATA") [31,32]. ILC3s are found in the greatest abundance at the mucosal barrier surfaces [27], and exhibit receptor-related orphan receptor (ROR) [33]. The so-called lymphoid tissue inducer (LTi) cells represent an ILC3 subtype. G-protein-coupled receptor 183 (GPR183) was expressed and migrated to its oxysterol ligand 7,25-hydroxycholesterol (7,25-OHC) in ILC3s with an LTi phenotype. In a murine experiment, ILC3s failed to localize to LAs and ILFs in mice missing Gpr183 or 7,25-OHC. A deficit in LA and ILF formation in the colon was induced by a Gpr183 deficiency in ILC3s. Colonic lymphoid tissue growth required localized oxysterol production by fibroblastic stromal cells, and inflammation-induced enhanced oxysterol production produced colitis via GPR183-mediated cell recruitment [34].

LTi cells are required for the formation of PPs and lymph node organogenesis, as well as the reorganization of lymphoid tissue following infections and the promotion of adaptive barrier immunity [27]. LTi cells can produce T helper 17 cell (Th17)-related cytokines (e.g., IL-17, IL-22) [35,36]. During homeostasis and mucosal inflammation, the commensal microbiota shapes the function of these cells [21].

## 4. Innervation and Vascularization of Isolated Lymphoid Follicles

According to murine and human studies, not only the normal gut mucosa—devoid of lymphoid follicles—is densely innervated, but so is the PP area [37,38]. ILFs/PPs are immune surveillance sites that are innervated within the gastrointestinal tract. The suprafollicular dome region contains a dense neural network, but not the germinal centers of PPs [37]. Extrinsic nociceptor neurons have recently been shown to mediate a protective immune response against *Salmonella* infection through direct sensing and release of calcitonin-gene-related peptide, which suppresses M-cell density and maintains segmental filamentous bacterial colonization of the ileum [38]. It has also been proposed that enteral sympathetic neurons are involved in the first stage of prions’ neuronal invasion, and that their associated neurons are potential prion carriers in the central nervous system [39,40].

Vasculogenesis plays a multifaceted role in mucosal organization; it is required for nutrition as well as metabolic processes. In terms of regeneration, bone-marrow-derived stem cells pass through vessels on their way to the site of tissue damage; they can promote malignant transformation by enabling metastasis formation.

The vessels beneath the mucosa shape the lymphatic network of ILFs; they form polyhedric meshes after surrounding each lymphoid follicle, and then encircle and completely envelop the nearby lymphoid follicles [14,41,42]. The tunica mucosa interstitial fluid enters the peripheral absorbent lymphatics in the deeper layers of the mucosa [14,41,42]. Blood vessels emerge from the arterioles beneath the mucosa, travel with the lymphatics to the tunica mucosa, and send thin branches to the germinal center of the ILFs; they then merge in a dense venous network and, eventually, continue via the peri-interfollicular venules [14,41,42]. In the vascular network, mucosal vascular addressin cell adhesion molecule (MAdCAM)-1-expressing and -non-expressing vessels are included [9].

Cytokines and intercellular adhesion molecules convert certain blood vessels to high endothelial venules (HEVs) during an inflammatory response [43]. Lymphocytes and neutrophils are thought to reach the site of inflammation via HEV-related trans- and intercellular pathways [43,44,45]. In a similar manner, bone-marrow-derived stem cells involved in regeneration could extend the site of mucosal damage as well [46].

## 5. Composition and Immune Function of Isolated Lymphoid Follicles

In humans, plasma cells and naive lymphocytes are restricted to ILFs and GALT-free LP, respectively, according to the novel isolation method proposed by Fenton et al. [15]. This finding suggests that ILFs serve as an inductive site of adaptive immunity. A total of 90% of the cells in ILFs are lymphocytes, with a slightly higher proportion of T cells than B cells. CD57^hi^PD-1^hi^ follicular T cells and CD4^+^CD25^+^CD127^-^IL2^-^ regulatory T cells are mostly found in ILFs, whereas cytokine-producing T-cell subsets are mainly located in the GALT-free LP. Furthermore, ILFs contain B cells expressing immunoglobulin (Ig)M, IgG, IgA1, and IgA2, indicating that B-cell class switching (a biological mechanism that changes a B cell’s production of immunoglobulin from one type to another) occurs within the germinal center of ILFs. ILFs are also a source of intestinal IgA molecules. A low overlap of IgA clones was detected in relation to PPs and SM-ILFs using IgA sequencing analysis, indicating a different profile within the intestinal immune response [15,47].

ILFs are made up of macrophages and dendritic cells that express APRIL (a proliferation-inducing ligand) and B cells that express TACI (transmembrane activator and calcium-modulating cyclophilin ligand interactor) [48,49]. ILF germinal centers contain follicular dendritic cells (FDCs) [50,51]. FDCs can keep native antigens as immune complexes on their membrane for months, before presenting them to B cells during the secondary response. FDCs protect bound B cells from apoptosis and promote B-cell differentiation into long-term memory B-cell clones [52]. Tfh cells, germinal centers, and FDCs were found in ILFs, along with CD40L and AID (activation-induced deaminase) enzyme expression, indicating that ILFs serve as a critical priming site for T-cell-dependent B-cell responses [15,48,49,53,54,55] (Figure 2).

## 6. Isolated Lymphoid Follicles in Colonic Inflammation

Colitis is a condition characterized by inflammation of the colonic epithelium and wall. Colitis is classified into specific types (inflammatory bowel disease (IBD), microscopic colitides, ischemic colitis, and infectious colitides) and non-specific types. The latter means that the colitis sample lacks pathognomonic histologic signs of IBD, the inflammatory process affects only the upper half of the mucous membrane, and it cannot be classified into distinct, specific forms.

Occupying a global health market, IBD—which includes ulcerative colitis (UC) and Crohn’s disease (CD)—is becoming increasingly prevalent worldwide. Genetic susceptibility, environmental variables, and microbiological factors all play roles in the development of IBD, which is defined as a chronic immune-mediated intestinal inflammation [56]. IBD has been attributed to excessive lymphocyte homing caused by increased expression of endothelial adhesion molecules. Historically, infiltrating lymphocytes have been defined as Th1 cells, Th17 cells, or B cells, all of which produce pro-inflammatory cytokines potentiating intestinal proteases, resulting in mucosal damage [57].

While secondary lymphoid organs (e.g., PPs, ILFs) develop at anatomically defined sites, chronic inflammation of peripheral tissues can result in the rapid formation of tertiary lymphoid organs (TLOs) [58] (Figure 3). Both the innate immune response against luminal microbiota and the lymphocyte function are impaired in IBD.

Recent discoveries have highlighted the role of ILCs in intestinal mucosal homeostasis and IBD [57,59,60,61,62]. Anomalies in the ILC population balance disrupt intestinal homeostasis, resulting in gut inflammation. The balance of ILC1s and ILC3s is closely linked to IBD and gut inflammation [19].

In CD patients, for example, the intraepithelial IFN-γ-producing ILC1 population is expanded in response to IL-12 and IL-15. This rise is accompanied by a decrease in natural cytotoxicity receptor (NCR)^+^ ILC3s, which worsens the disease. In the lamina propria of inflamed tissues from CD patients, the ILC population is skewed toward the CD127^+^ ILC1 subset [19,63]. In response to IL-12 and IL-18, these CD127+ ILC1s produce a large amount of IFN-γ [19].

Gut NK cells, like ILC1s, control infection, but they can induce inflammation and IBD [64]. Inflammation in the intestine is linked to a high level of endoplasmic reticulum stress in intestinal epithelial cells. Endoplasmic reticulum stress in intestinal epithelial cells is exacerbated by the increased expression of natural killer group 2 member D in NK cells, which recognize and destroy stressed intestinal cells [65]. CD patients have an increased NK cell population in the lamina propria, which facilitates the proliferation of CD4^+^ T cells as well as Th17 development [19]. The balance of NKp44^+^ and NKp46^+^ NK cells in the intestinal mucosa of CD patients is disrupted. CD may be mediated by NKp46^+^ NK cells producing IFN-γ, which activates intestinal inflammatory macrophages in CD patients’ intestinal mucosa [19]. In primary sclerosing cholangitis, which is an idiopathic chronic hepatobiliary system disorder linked to UC, an increase in the number of IFN-producing ILC1s was also demonstrated [66].

In IBD, not all ILC3 populations are reduced. Some ILC3 cytokines, transcription factors, and cytokine receptors—such as IL-12A, IL-22, RORC (RAR-related orphan receptor gamma), AHR (aryl hydrocarbon receptor), and IL-23R—have increased gene expression in ILCs from CD patients [19,67]. ILC3 expression of IL-17 and IL-22 is induced by IL-23 [19]. Human MALT has also been found to contain IL-23-reactive ILC3s [19]. CD patients, unlike UC patients, have elevated expression of IL-17A and IL-17F in mucosal ILCs [19]. ILCs in the intestine—including ILC3s—enter and exit cryptopatches in a highly dynamic manner. ILC3s mobilize from CPs during colitis, and initiate inflammatory immune cascades that result in intestinal inflammation [19].

There are fewer NCR^+^ ILC3s generating IL-22 in the mucosa of CD patients, which may affect barrier integrity and contribute to the development of IBD [19]. RegIIIγ, a regenerating islet-derived protein, is also reduced in people with IBD. Lower levels of regenerating islet-derived protein (Reg)III, RegIIIb, alpha-1, 2-L-fucosyltransferase (Fut)2, and mucin protein (Muc2)—as well as reduced expression of the tight junction protein claudin-2 and poor intestinal epithelial cell regeneration—are also found in IBD patients [19,68]. ILC3s display a high level of the TL1A receptor, also known as death receptor 3 (DR3). DR3 antibody ligation aggravates colitis by stimulating ILC3 granulocyte–macrophage colony-stimulating factor production via the p38 MAPK (mitogen-activated protein kinase) pathway, resulting in the accumulation of CD11b^+^ CD11c^+^ myeloid cells and the facilitation of ILC3 loss from the intestine via an IL-23-dependent mechanism. As a result, antibody-mediated DR3 blockade alleviates colitis, possibly by suppressing the harmful effects of ILC3s [69]. In summary, NCR^+^ ILC3s are reduced in IBD, despite their importance in maintaining intestinal homeostasis by producing IL-22, whereas IL-17-producing NCR^+^ ILC3s contribute to disease development.

The role of ILC2s in IBD is unknown. A possible proinflammatory role for ILC2s has been suggested in UC patients with inflamed mucosa, as their levels are higher than those found in healthy persons or non-inflamed regions [63]. Elevated signaling levels of lymphocytic activation molecule family member 1 (SLAMF1) and human leukocyte antigens (HLA-)DR indicate ILC2 activation in CD patient blood [19]. In CD patients with active illness, the frequency of SLAMF1+ ILC2s correlates inversely with disease severity, and SLAMF1^+^ ILC2s appear to represent a mature population that produces IL-13 and high amounts of prostaglandin D2 receptor 2 (PTGDR2/CRTH2), CD161, and GATA3. [70].

Transmural inflammation can affect any site of the gastrointestinal tract in CD, but the disease most commonly affects the ileum and the proximal colon [71]. Aphthous ulcers—inflammatory lesions of the FAE of PPs and ILFs—are among the first visible signs of CD [72,73], implying that ILFs may play a crucial role during the onset of CD. TLOs are frequently found around occluded lymphatic vessels in the LP, and also around aphthous ulcers in CD [74,75]. However, TLOs may be present in the deeper layers of the small and large bowel, including muscle and mesenteric adipose tissue, because transmural inflammatory infiltrates are characteristic of CD [76].

In UC, the inflammation of the mucosal layer starts in the rectum and spreads along the colon, rarely progressing into the terminal ileum [77]. In UC, the inflamed colon is enriched in cells and molecules associated with organized lymphoid structures, such as HEVs, Tregs, central memory T cells, follicular B cells, LA-associated stromal cells, and C–C motif chemokine ligand 19 (CCL19) [78,79,80,81]. Although the number of visible LAs in UC-affected colonic tissue is increased [9,82], it is still unclear whether these are enlarged ILFs or inflammation-induced TLOs.

Recently, in murine DSS-induced colitis, it has been demonstrated that mice develop severe colitis in addition to rapid TLO neogenesis due to failed antimicrobial immunity (i.e., RORt/mice; lacking ILC3 and Th17 responses) [83]. Antibiotic treatment reverses this pathology and TLO formation, confirming the pathology’s reliance on a commensal microbial insult [83]. As a result, the presence of TLOs during chronic intestinal inflammation clearly indicates a type of organ-specific immune response dysregulation.

The development of extraintestinal manifestations (EIMs) during the course of IBD is a well-known clinical phenomenon [84,85]. While TLOs are more common in CD than in UC, EIMs are more frequent in CD patients [84]. The temporal onset of arthritis correlates with luminal dysbiosis and the development of intestinal TLOs in a TNF-transgenic model of CD and polyarthritis [86,87]. This study provides experimental evidence that abnormal antibody formation in intestinal TLOs may occur as a result of immune dysregulation, promoting the emergence of EIMs.

## 7. Isolated Lymphoid Follicles and Tertiary Lymphoid Organs in Colorectal Carcinogenesis

Colorectal cancer (CRC) is one of the most common types of cancer, and is one of the leading causes of cancer-related death worldwide [17]. Classic CRC is a malignant disease that is caused by a variety of factors, including genetic mutations, epigenetic changes, chronic inflammation, diet, and lifestyle. The molecular mechanisms underlying CRC tumorigenesis and progression, however, are not completely understood. Accumulating evidence suggests that the gut microbiota contributes to the development of colorectal cancer, in part through its interactions with the subepithelial compartment [17].

Tumor-infiltrating T cells play an important role in the clinical outcome of CRC, but the dynamics of their recruitment during CRC’s clinical progression is unknown. In CRC, the Crohn’s-like lymphoid reaction (CLR) corresponds to a peritumoral lymphoid aggregate located at the tumor’s advancing edge. The CLR has a similar cellular composition and structure to secondary lymphoid organs, implying that it is a functional immunological component contributing to the adaptive immune response in CRC [88]. CLRs are TLOs that are specific to CRCs; they are structurally and functionally similar to other cancer-related tertiary lymphoid formations, but with distinct characteristics and clinical benefits [88]. CLRs are distinguished immunologically by the increased intratumoral lymphocyte infiltration [89,90]; this link appears to be independent of CRC’s microsatellite instability status [91].

According to multivariate statistical analyses, CLRs have a greater impact on survival prediction than traditional histological parameters [91,92,93], and tumor-infiltrating lymphocytes and CLRs provide more prognostic information than the TNM classification in early-stage CRC (N0M0) [91]. CLRs have been shown to predict responses to chemotherapy in metastatic CRC, implying that cytotoxic chemotherapy may have a synergistic effect on the adaptive immune response to metastatic disease [94,95]. CLRs have continuously been correlated with favorable prognostic variables and enhanced survival in patients with CRC, frequently offering more information about prognosis than current clinical staging methods.

Antitumor immunity is thought to be boosted by B cells infiltrating tumors [96,97,98,99]. B-cell subgroups are quite diverse, with distinct roles and diverse modes of action. Because B cells form TLOs that can perform multiple functions, B-cell biology, adaptive immunity, inflammation, and the tumor microenvironment are all intertwined. These are local (and systemic) antitumor response centers where various cell populations of the B-cell line, Tfh cells, FDCs, and HEVs congregate and act in concert [50,100,101,102,103].

TLO microarchitecture is strikingly similar to the classical germinal centers of secondary lymphoid organs. The active TLO is AID (activation-induced cytidine deaminase)-positive, and functions as a regulator of class-switching recombination and somatic hypermutation of immunoglobulin genes [104,105]. TLO activity products, such as tumor-commanded plasma cells or memory B cells, may have a direct effect on cancer pathology. Not only are high-affinity antibodies of various isotypes produced against local tumor antigens, but TLO-associated cytokines are also secreted, which coordinate immune and stromal cells in the tumor’s immune microenvironment. Furthermore, memory B cells migrate through the bloodstream to distant tissues, including metastases [106,107,108].

TLOs appear to be physiologically present in certain tissues, such as the gastrointestinal mucosa, and play a role in balancing the protective immune response and self-tolerance [109]. The anatomical features, cell composition, activation, and proliferation status of ILFs were characterized and compared using a panel of lineage marker immunostainings (e.g., CD20, AID, PNAd, CD27, CD138, CD3, Ki67, CD208), and the clinical relevance of datasets was assessed [100,110,111,112,113]. It has been proposed that there is a link between the immune phenotype of ILFs in non-tumorous colon tissue and the lymphoid structures that develop at primary and metastatic sites. It has been demonstrated that the immunophenotype of TLOs in CRC and metastatic CRC is predetermined by the characteristics of ILFs from the non-tumorous colon. In terms of clinical significance, it was discovered that ILF characteristics could encrypt information on disease outcome. ILFs and TLOs that are B-cell-enriched and highly proliferating may indicate a better clinical outcome. The ILF-based assessment adds significantly to the knowledge gained from the Immunoscore analysis [114]. The data also revealed a link between non-tumorous colonic B cell clonality and the liver’s metastatic tumor features. B cells were identified as essential components of ILFs and TLOs. The B-cell-enriched immune phenotype of ILFs within non-tumorous tissue is associated with improved prognosis of CRC in liver metastasis patients, and the variable characterizing the magnitude of CD20-positive cells within ILFs can be used as a marker to stratify patients into risk groups. It was also emphasized that CD20-positive B cells are exclusively localized within ILFs in non-tumorous tissue, as opposed to the heterogeneous T-cell distribution pattern that is not associated with ILFs. As a result, CD20 is a highly specific marker for lymphoid structures within the colonic mucosa [100].

CD27 is a marker of memory B cells, plasma cells, and subsets of memory T cells [100], and it can characterize the outcome of the activity of the germinal center reaction. The vast majority of cells from various lineages within ILFs are CD27-positive [100]. This finding suggests that this regulatory molecule plays an important role in the biology, maintenance, and functionality of ILFs in the colon. These CD27-related discoveries [100] also highlight this molecule as a novel target for modulating ILF-driven antitumor immune responses. A potential future research question is whether agonistic anti-CD27 antibodies [115,116] can increase the intrinsic power of ILFs in the colon.

Patients with IBD are at an increased risk of developing CRC, which is thought to be caused by persistent intestinal inflammation. ILC3 has been shown to infiltrate tumors in a variety of tissues, and has been associated with the onset, progression, and control of carcinogenesis in colorectal cancer—a result backed by an increasing body of experimental evidence from murine colon cancer models [117]. ILC3 activation and cytokine production appear to be critical to their impact on colorectal cancer. In human colon tumors, IL-23 expression is higher than in healthy tissue, and it has been linked to a poor prognosis and more aggressive disease [118]. Additionally, gene polymorphisms associated with IBD and colitis-associated cancer (CAC)—such as *Card9*—are essential for the regulation of ILC3 activation and IL-22 production via the ILC3-activating cytokine IL-1 [119]. These data imply that deregulation of the ILC3 and/or Th17 axes is critical for the development and progression of colorectal cancer. As a result, ILC3-associated cytokines have been implicated in the pathogenesis of CRC, but have been assigned inconsistent roles, highlighting the intricacies of ILC3 biology in health and disease. For instance, both IL-17A and IL-22 have been associated with human colorectal cancer, and the increase in IL-17-producing cells has been proven to be an independent prognostic marker in human colorectal cancer [119]. Importantly, polymorphisms in IL-22 have been linked to an increased risk of developing CRC [119].

With regard to the microbiome, it has been demonstrated that the different anatomical localization affects the bacterial flora in the gastrointestinal tract both longitudinally and cross-sectionally. The colonic microbiota influences the progression of CRC through several mechanisms, such as mucosal inflammation, dysbiosis, and bacterial metabolites [17]. It is notable that colonic commensal bacterial translocation is also linked to CRC progression and tumorous cachexia [17].

## 8. Future Perspectives

Recent findings have significantly increased our understanding of GALT heterogeneity, composition, and immune function [9]. ILFs are common in pediatric intestinal biopsies, and their hypertrophy has been linked to allergic sensitization [120]. Because ILFs are also the primary source of soluble IgA, their potential actions in the induction of immune tolerance and as a therapeutic target are highlighted [15].

There is compelling evidence that ILCs play a role in tissue regeneration [21]. Increased ILC3-derived IL-22 levels in patients with acute IBD are correlated with mucosal healing—a key clinical endpoint in IBD [121]. The development of ILC-targeting molecules may further support the conventional treatment options for human colitides.

The prognosis of advanced, metastatic CRC is definitely the worst. A novel monoclonal antibody (MHA112) against peripheral node addressin (PNAd) has recently been developed [122]. PNAd is a glycoprotein expressed in HEVs, and is constitutively present in lymph nodes and formed ectopically in CLRs and TLOs, as well as in human metastatic lesions [122,123]. The MHA112 drug (Taxol)-delivery platform simultaneously targets the three key cancer sites (primary tumor, tumor-draining lymph nodes, and metastatic niches) and, thus, appears to be a promising, novel therapy for CRC. The growing pipeline of monoclonal antibody therapies as well as small-molecule agonists and antagonists that target the related Th17 pathway may allow ILC3-targeting therapeutics to enter the clinic more quickly, raising awareness of the cells’ importance not only in the treatment of chronic inflammatory disease, but also in inflammation-associated cancer.

Further research will most likely lead to better understanding of how immune responses are initiated and controlled within the large intestine, thus leading potentially to the development of novel mucosal vaccination and treatment strategies critical for the management of IBD and CRC.

## Figures and Tables

**Figure 1 biomedicines-10-00226-f001:**
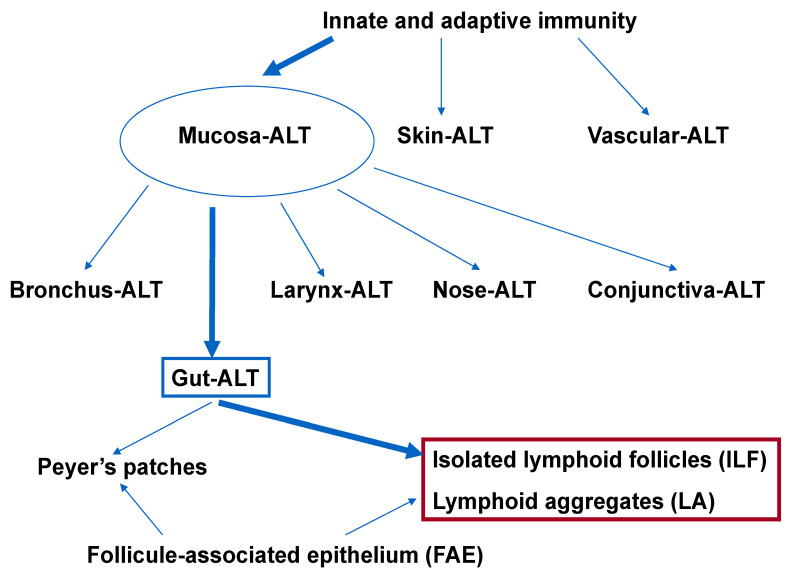
Organization of the gastrointestinal lymphoid tissue: Nearly 70% of innate and adaptive immune cells are found in mucosal-associated lymphoid tissue. In addition to MALT, SALT and VALT also exist. One part of MALT is GALT, whose major lymphoid organs are PPs, ILFs, and Las; each has an FAE with a distinctive structure and function. The highlighted components refer to the colon. ALT: associated lymphoid tissue.

**Figure 2 biomedicines-10-00226-f002:**
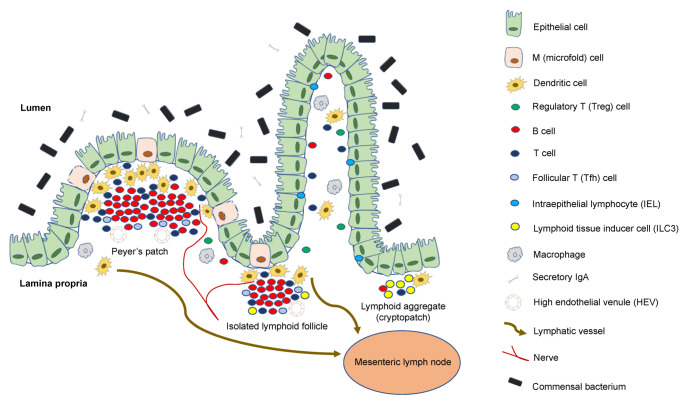
Composition of Peyer’s patches and isolated lymphoid follicles: The intestinal lamina propria is separated from the gut flora by a single epithelial cell layer; it is protected by a thick layer of mucus, bactericidal defensins, neutrophils, and a large amount of secretory IgA specific to antigens. Intraepithelial lymphocytes monitor epithelial damage, and may recognize microbial antigens within the epithelium. The lamina propria is densely packed with T cells, plasma cells that produce IgA, and macrophages; it also contains a large number of dendritic cells, which migrate through the lymph to mesenteric lymph nodes, and present antigens to T cells. Microfold cells are found in the follicle-associated epithelium, and transport luminal antigens to Peyer’s patches or isolated lymphoid follicles for collection by dendritic cells, which can also collect antigens from apoptotic epithelial cells.

**Figure 3 biomedicines-10-00226-f003:**
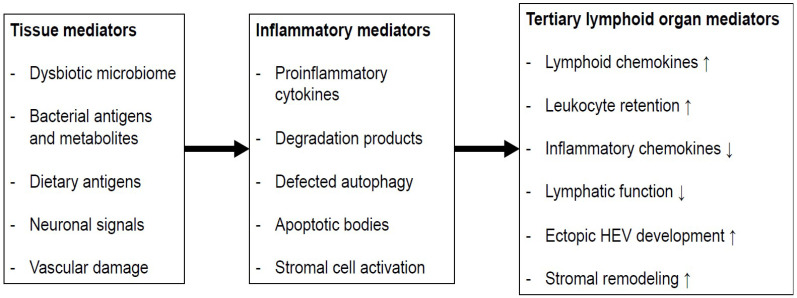
The putative process of intestinal TLO formation: Tissue factors present in the colon may result in the release of inflammatory mediators that enable the processes that promote the development of TLOs. HEV: high endothelial venule; ↑: increase; ↓: decrease.

## Data Availability

Not applicable.

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
