# Peer review of "Colitis and Colorectal Carcinogenesis: The Focus on Isolated Lymphoid Follicles"

_biomedicines, 2022, doi:10.3390/biomedicines10020226_

Round 1

Reviewer 1 Report

Györgyi Műzes et al demonstrated that the importance of lymphoid follicles in colon.

I have some comments.

The title includes colorectal cancer, but there is little information on colorectal cancer. The author should have explained colorectal cancer more clearly. In particular, whether the system is tumor promotive or tumor inhibitory to colorectal cancer should be clearly explained.

This paper has the same problem with colitis.

There is a short description of IBD, but no description of colonic inflammation in general.

It is difficult to understand whether the references in the paper are studies using clinical tissues or animal experiments. Since the meanings are very different, the discussion should clearly specify whether the results are from human or animal studies.

The word microbiome is used a lot, but it is necessary to be more specific about what it refers to.

In the section on conclusion, there are many topics that the author have not discussed before. In the conclusion, the author should summarize the previous contents and write a conclusion.

The section on ILC in the first part of the manuscript is long and seems to be the main focus of the manuscript. The title should be revised.

The overall structure should be revised because it is not well organized and it is not clear what the author wants to talk about.

Author Response

Dear Reviewer,

We appreciate your constructive criticism, helpful advice, and insightful comments. Our answers to your comments are summarized below.

# 1. „The title includes colorectal cancer, but there is little information on colorectal cancer. The author should have explained colorectal cancer more clearly. In particular, whether the system is tumor promotive or tumor inhibitory to colorectal cancer should be clearly explained.”

Our answer: We briefly describe the basic characteristics of colorectal cancer in Chapter 7, paragraph 1.

# 2. „This paper has the same problem with colitis. There is a short description of IBD, but no description of colonic inflammation in general”

Our answer: We briefly deal with the basic characteristics of colonic inflammation and IBD in Chapter 6, paragraph 1.

# 3. „It is difficult to understand whether the references in the paper are studies using clinical tissues or animal experiences. Since the meanings are very different, the discussion should be clearly specify whether the results are from human or animal studies.”

Our answer: In the first paragraph of Chapter 5, we indicate data of human origin. Chapter 6 is about human data as well (which is obvious from the text). The penultimate paragraph of Chapter 6 is about a murine model, and this fact is also clearly indicated. Data reported on colon cancer shows human abnormalities in both sporadic CRC and colitis-associated cancer. This is obvious from the text described that refers to human disease conditions.

# 4. „The word microbiome is used a lot, but it is necessary to be more specific about what it refers to.”

Our answer: Between lines 78-95, we have described the microbiome related to ILFs.

# 5. „In the section on conclusion, there are many topics that the authors have not discussed before. In the conclusion, the author should summarize the previous contents and write a conclusion.”

Our answer: The final chapter's title has been ruined. We did not intend to conclude the last chapter at the conclusion of a summary article, but rather to raise issues that may provide direction for future questions. As a result, the chapter's title has been changed to “Future Perspectives”.

# 6. „The section on ILC in the first part of the manuscript is long and seems to be the main focus of the manuscript. The title should be revised.”

Our answer: Chapter of "The innate lymphoid cell" is only a total of 410 words long. In 1065 words, the role of ILFs in inflammation was described, and in 1062 words, the role of ILFs in colon cancer development. Omitting ILCs would be a professional error, as these relatively recently discovered cells play a critical role in the formation of both the colonic epithelium and secondary lymphoid structures of the colon. ILCs are involved in physiological development and pathological processes such as inflammation and carcinogenesis as well. On this basis, we believe that this review can hardly be without a brief but concise description of ILCs. The article, however, primarily focuses on the role of ILFs in the development of pathological conditions, and in this respect, ILCs represent an important subset.

# 7. „The overall structure should be revised because it is not well organized and it is not clear what the author wants to talk about.”

Our answer: The purpose of this review article is to demonstrate the essential role of secondary lymphoid organs in inflammation and carcinogenesis of the colon. Probably not all readers are familiar enough with the details of the main topic. Accordingly, the first section of the article deals briefly with the organization of the intestine's secondary lymphoid organs, their relationship to luminal microorganisms, the main characteristics of cells involved in the formation and development of lymphoid structures (ILC chapter), and the anatomy and cellular composition of lymphoid organs. In the first, shorter half of the article, these topics are discussed. In the following, based on recent findings, the role of ILFs as functional units in inflammation and carcinogenesis is analyzed. In our opinion, the present format of the article enables the readers to perceive the subject.

We sincerely hope you accept the corrections to the paper you've suggested along with the answers to your questions, and that this revised article is already acceptable for publication.

Reviewer 2 Report

The authors present a very interesting review on Colitis and Colorectal Cancer focusing on Isolated Lymphoid Follicles (ILF). The authors cited numerous peer reviewed articles to enrich the article and make it up to date. However, there are several concerns for general readers.

Major comments

  1. In pages 61- 66; although Figure 1, describes the organization of gastrointestinal lymphoid tissue, it must require a graphical cartoon to show the location of different lymphoid follicles.

Without the cartoon the review itself stands short of perfection.

  1. In pages 83 – 87, please name the specific commensal bacteria.

  1. In pages 155 -160, please concise and simplify the composition of ILF for general reader. Also explain “class Switching” in simple words.

  1. In page 280, please explain “MSI”.
  2. In pages 304 - 306, give some examples of lineage markers.
  3. In pages 327, please elaborate and complete this incomplete sentence.

Author Response

Dear Reviewer,

We appreciate your constructive criticism, helpful suggestions, and insightful comments.

Our answers to your comments are summarized below.

# 1. „In pages 61-66; although Figure 1. describes the organization of gastrointestinal lymphoid tissue, it must require a graphical cartoon to show the location of different lymphoid follicles. Without the cartoon the review itself stands short of perfection.”

Our answer: We prepared an overview figure that clearly indicates the location and composition of each lymphoid structure.

# 2. „In pages 83-87, please name the specific commensal bacteria”

Our answer: Between lines 78-95, we describe the microbiome related to ILFs, as well as indicate the names of commensal bacteria.

# 3. „In pages 155-160, please concise and simplify the composition of ILF for general reader. Also explain „class switching” in simple words.”

Our answer: Between lines 165-172, we simplified the text and explain the term „class switching”.

# 4. „In page 280, please explain „MSI”.”

Our answer: MSI stands for microsatellite instability (we indicated it in lines 319-320).

# 5. „In pages 304-306, give some examples of lineage markers.”

Our answer: In line 349 we list the used lineage markers.

# 6. „In page 327, please elaborate and complete this incomplete sentence.”

Our answer: Sorry for the mistake, the incomplete sentence has been canceled.

We sincerely hope you accept the corrections to the paper you've suggested, and that this revised article is already acceptable for publication.

Round 2

Reviewer 1 Report

The author responded sincerely to the reviewers' questions and improved the manuscript.